# EMERGENT DEXTERITY VIA DIVERSE RESETS AND LARGE-SCALE REINFORCEMENT LEARNING

**Patrick Yin**[1,*]    **Tyler Westenbroek**[1,*]    **Zhengyu Zhang**[2]    **Ignacio Dagnino**[1]

**Eeshani Shilamkar**[1]    **Numfor Mbiziwo-Tiapo**[1]    **Simran Bagaria**[3]    **Xinlei Liu**[1]

**Galen Mullins**[3]    **Andrey Kolobov**[3]    **Abhishek Gupta**[1]

[1]University of Washington    [2]NVIDIA    [3]Microsoft Research    *Equal contribution

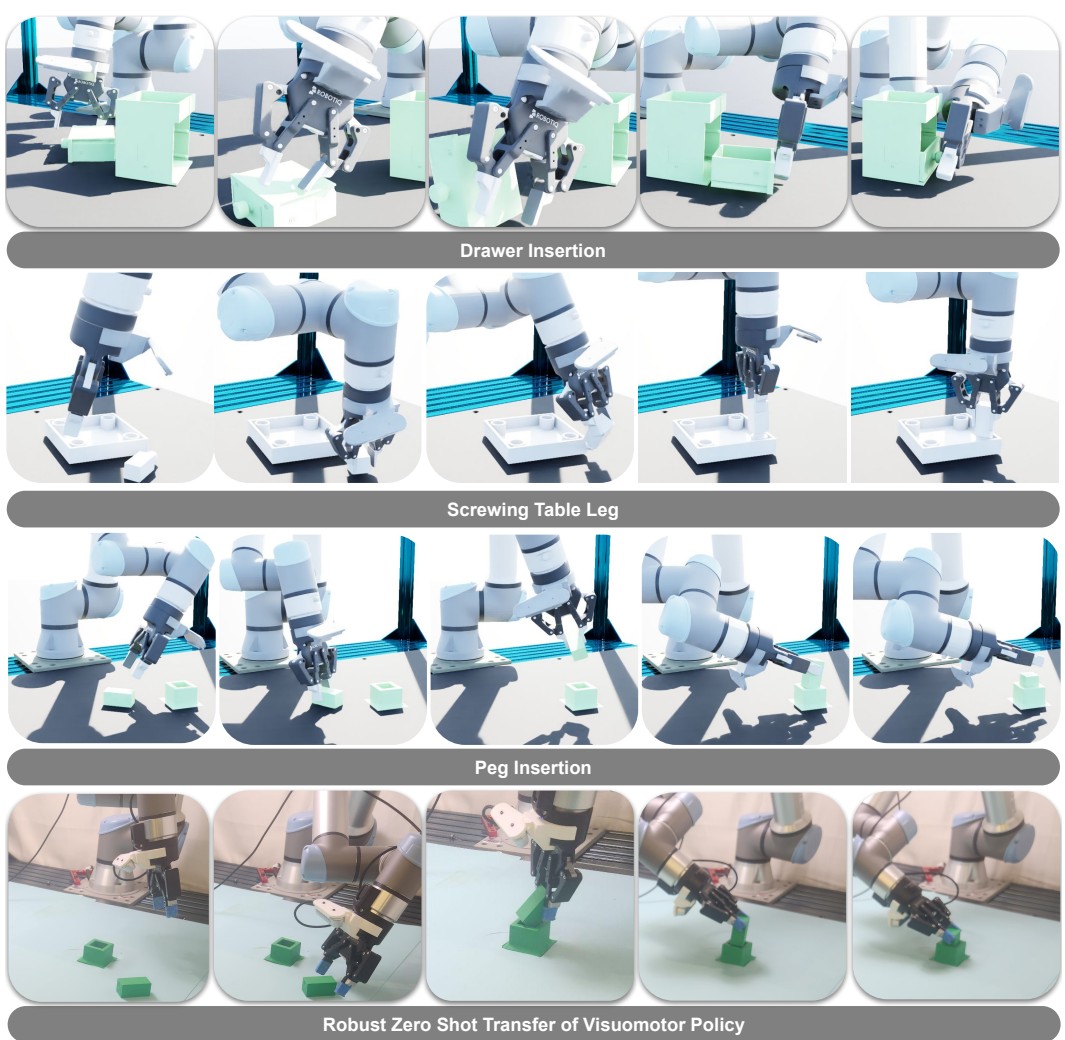

**Drawer Insertion**

**Screwing Table Leg**

**Peg Insertion**

**Robust Zero Shot Transfer of Visuomotor Policy**

Figure 1: `OmniReset` automatically generates diverse reset states, enabling complex, multi-phase manipulation behaviors to emerge from large-scale reinforcement learning. Using the same generic reset procedure for each task, `OmniReset` learns robust, task-specific policies. The first row shows the robot pushing and flipping a drawer before wiggling it in. The second row shows the robot picking up a table leg, adjusting its grip using the table, and twisting the leg into the hole. The final two rows show peg insertion in simulation and a distilled RGB policy attempting the task on a real robot, which recovers from a failed insertion to successfully complete it, demonstrating robust emergent retrying behavior.

## ABSTRACT

Reinforcement learning in massively parallel physics simulations has driven major progress in sim-to-real robot learning. However, current approaches remain brittle and task-specific, relying on extensive per-task engineering to design rewards, curricula, and demonstrations. Even with this engineering, they often fail on long-horizon, contact-rich manipulation tasks and do not meaningfully scale with compute, as performance quickly saturates when training revisits the same narrow regions of state space. We introduce `OmniReset`, a simple and scalable framework that enables on-policy reinforcement learning to robustly solve a broad class of dexterous manipulation tasks using a single reward function, fixed algorithm hyperparameters, no curricula, and no human demonstrations. Our key insight is that long-horizon exploration can be dramatically simplified by using simulator resets to systematically expose the RL algorithm to the diverse set of robot-object interactions which underlie dexterous manipulation. `OmniReset` programmatically generates such resets with minimal human input, converting additional compute directly into broader behavioral coverage and continued performance gains. We show that `OmniReset` gracefully scales to long-horizon dexterous manipulation tasks beyond the capabilities of existing approaches and is able to learn robust policies over significantly wider ranges of initial conditions than baselines. Finally, we distill `OmniReset` into visuomotor policies which display robust retrying behavior and substantially higher success rates than baselines when transferred to the real world zero-shot. Project webpage: https://omnireset.github.io

## 1 INTRODUCTION

Reinforcement learning (RL) in massively parallelized simulation environments (Mittal et al., 2023; Todorov et al., 2012) has driven recent successes in sim-to-real robotics (Akkaya et al., 2019; Hwangbo et al., 2019). In principle, these algorithms can automatically acquire complex, contact-rich behaviors through repeated interaction with the environment. Yet in practice, obtaining robust and performant policies still requires extensive per-task engineering, limiting the scalability of this paradigm—particularly for long-horizon manipulation tasks. This sharply contrasts with domains such as language modeling, where simply scaling data and compute has yielded dramatic gains with similarly simple RL algorithms Guo et al. (2025). How can we achieve comparable scalability for robotic manipulation?

The central bottleneck in robotic RL is that standard exploration techniques Schulman et al. (2017); Haarnoja et al. (2018) saturate as parallelism and compute are scaled, repeatedly sampling a narrow state–action distribution and becoming trapped in local minima Singla et al. (2024). While numerous advanced exploration methods have been proposed Pathak et al. (2017); Burda et al. (2018), their added algorithmic complexity makes them difficult to scale in practice. As a result, practitioners often rely on human intuition to reduce the exploration burden, injecting structure through task-specific reward design Westenbroek et al. (2022); Ng et al., hand-designed curricula Lee et al. (2020a), or user-provided demonstrations Bauza et al. (2025); Peng et al. (2018); Nair et al. (2018). Although effective in many settings, these approaches are fundamentally limited by the amount of human effort they require. A complementary strategy is to simplify the learning problem through additional system scaffolding—using RL only for contact-rich phases while relying on motion planning or trajectory optimization for the remainder Lee et al. (2020b); Tang et al. (2024). While this reduces exploration demands, it comes at the cost of increased system complexity. These approaches reflect a prevailing assumption in robotics: that dexterous manipulation is too complex to emerge from large-scale RL alone and must instead be scaffolded with additional task-specific structure.

In contrast, we argue for *systematically exposing RL to a superset of the interactions it will encounter when manipulating the scene*, and then allowing dexterous task-specific behaviors to emerge from large-scale compute and optimization. Although the space of possible behaviors is vast—flipping, screwing, insertion, and other contact-rich motions—successful policies reuse a relatively small set of recurring interaction modes, such as approaching objects, initiating contact, and forming stable grasps. These modes can be densely covered through generic resets that do not encode task-specific solutions. By sufficiently randomizing object poses and sampling these interaction states,

we substantially reduce the exploration burden and ensure the agent encounters meaningful success signals. This coverage allows rewards to propagate smoothly through the state space, enabling the agent to identify high-value regions and learn how to stitch together multiple distinct behaviors to reach these goals, without task-specific shaping our guidance.

Concretely, we introduce `OmniReset`, a scalable framework for robotic manipulation that automatically generates diverse initial-state distributions to densely cover the contact-rich interactions the robot may encounter. This coverage allows PPO Schulman et al. (2017) to fully leverage large-scale compute, improving performance as the number of parallel environments increases. When scaled sufficiently, `OmniReset` learns to combine multiple interaction modes—such as pushing, flipping, and insertion (Fig. 1)—into coherent, multi-stage strategies without task-specific reward shaping, curricula, or demonstrations. Across diverse contact-rich tasks, this approach solves long-horizon problems that are out of reach for existing methods, producing robust policies that succeed from a wide range of initial states rather than narrow distributions typical of baseline approaches. Finally, we demonstrate that `OmniReset` can be used to train visuomotor policies via student-teacher distillation which robustly transfer to the real-world and substantially outperform real-world baselines.

## 2 RELATED WORK

**Exploiting Resets and in Reinforcement Learning**: Exploiting simulator resets for RL is a natural idea which has been explored in many contexts. Prior theoretical works (Kakade & Langford, 2002) have suggested more uniform sampling over initial states, but do not provide practical algorithms. The primary focus of many works is to make learning more tractable by generating an explicit curriculum over resets(Tang et al., 2023; Dennis et al., 2020; Bauza et al., 2025), for instance through a "reverse-curriculum" of states going backwards from the goal (Florensa et al., 2017) or using a learned dynamics model to propose viable resets (Edwards et al., 2018; Ivanovic et al., 2019). In contrast, a second category of methods leverage *demonstrations* (whether human or automatically generated) to generate feasible pathways to the goal (Tao et al., 2024; Resnick et al., 2018; Salimans & Chen, 2018; Bauza et al., 2025; Tang et al., 2024). In contrast to these prior works, we show that neither human demonstrations, nor a curriculum is needed, but rather that the simple approach for generating diverse resets in `OmniReset` naturally scales to various long-horizon manipulation problems without added algorithmic complexity.

**Exploration Strategies for Reinforcement Learning**: RL practitioners have designed a variety of exploration strategies to effectively uncover goal-reaching paths for a *fixed set of initial conditions*, with uninformative rewards. A major line of work is bonus-based exploration, where agents receive intrinsic rewards for visiting novel or unpredictable states. Count-based methods reward visits to rarely seen states (Ostrovski et al., 2017; Bellemare et al., 2016; Martin et al., 2017), while curiosity-based methods provide bonuses based on prediction errors (Burda et al., 2018; Sancaktar et al., 2022; Pathak et al., 2017). Other approaches (Osband et al., 2016; 2019; Russo et al., 2018) promote temporally correlated exploration by injecting stochasticity at the policy or value-function level. Finally, diversity-driven methods optimize for behavioral variety (Eysenbach et al., 2019; Rajeswar et al., 2023). These approaches are complimentary to our work; our main contribution is demonstrating that large scale-scale parallelization and resetting schemes lead to the emergence of surprising levels of dexterity without the need for advanced exploration incentives.

**Leveraging Demonstrations**: An alternative is to increasingly rely on human demonstrations and imitation learning to overcome difficult long-horizon exploration. Approaches include adding auxiliary BC loss terms to RL objectives (Nair et al., 2018; Hester et al., 2018; Rajeswaran et al., 2018), simply adding demonstrations to the replay buffer (Vecerik et al., 2017), and introducing reward shaping terms which encourage RL agent to follow demonstrations (Tang et al., 2024; Reddy et al., 2019; Koprulu et al., 2024; Peng et al., 2018). Other works have sought to squeeze more information out demonstrations by automatically translating existing demonstrations to new initial conditions and scenes (Mandlekar et al., 2023) or by robustifying BC policies by promoting recovery behavior (Ke et al., 2023; Ankile et al., 2024). While our work complementarily pushes the limits of what behaviors can be learned entirely from scratch, we expect that demonstrations (when available) can also be incorporated into our framework to further accelerate learning.

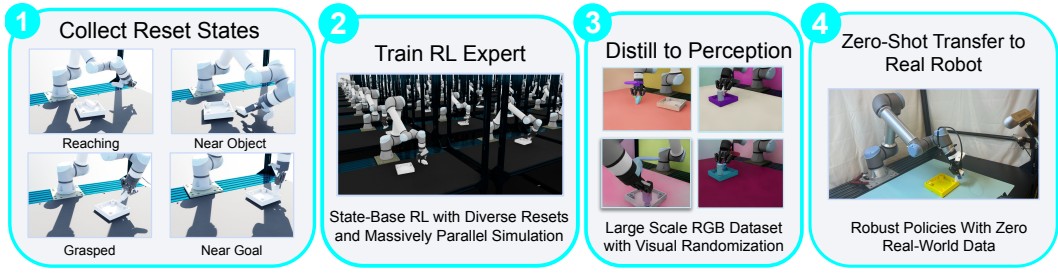

Figure 2: **Sim-to-Real Pipeline with `OmniReset`** (1) After generating partial assemblies and grasps from the simulator, (2) we collect reset states: reaching, near object, grasped, and near goal. (3) We then train a state-based RL policy initialized from these reset states, which is used to (4) train student-teacher distillation to get a RGB policy. (5) By finetuning this RGB policy on a mix of simulation data and small set of real demonstrations, (6) we deploy the RGB-based policy in the real world.

# 3 GENERATING DIVERSE RESETS FOR LEARNING DEXTEROUS MANIPULATION

This section introduces `OmniReset`, a scalable framework for robotic manipulation that automatically constructs RL problems by generating diverse, manipulation-centric reset distributions. Rather than relying on task-specific curricula, demonstrations, or carefully tuned rewards to guide exploration, `OmniReset` exposes standard RL algorithms to a broad superset of key interaction states that would be rarely encountered under naïve exploration. When paired with large-scale simulation and compute, this reset design continually exposes the algorithm to diverse interaction states, preventing convergence to narrow, suboptimal behaviors and enabling complex multi-step manipulation skills to emerge from large-scale optimization.

## 3.1 PROBLEM SETTING

**Reinforcement Learning Problem:** We formalize the RL problems synthesized by `OmniReset` as a Markov decision process (MDP) defined by the tuple $(\mathcal{S}, \mathcal{A}, P, r, \gamma, \rho)$, where $s \in \mathcal{S}$ denotes the state, $a \in \mathcal{A}$ denotes the robot's action, $s' \sim P(\cdot|s,a)$ denotes the next state sampled from the transition dynamics, $r$ is the reward function, $\gamma$ is the discount factor, and $s_0 \sim \rho$ is the initial state distribution. We optimize for the discounted sum of rewards: $J(\pi) = \mathbb{E}_{s_0 \sim \rho, a \sim \pi}[\sum_{t=0}^{\infty} \gamma^t r(s_t, a_t)]$, where $a \sim \pi(\cdot|s)$ denotes actions sampled from the policy. We focus our RL training on compact state representations (i.e., Lagrangian states), only moving to vision-based distillation for transfer to the real world (Sec. 5).

**Task Scope and User Inputs:** To effectively leverage task structure when automatically designing RL problems we make several practical assumptions. First, we focus on manipulating rigid bodies where the goal is to move a single user-specified object to a target configuration (potentially relative to other objects). For example, the assembly task in Figure 1 requires picking up a table leg, moving it to the desired hole, and screwing the pieces together—a long-horizon task requiring diverse behaviors, yet framed as moving one object to a goal. Specifically, `OmniReset` requires the following input from the user:

**Requirement 1** *A target object $s^{tar} \subset s$ to be manipulated.*

**Requirement 2** *A set of goal configurations $\mathcal{G} \subset \mathcal{S}$ for $s^{tar}$.*

**Requirement 3** *A workspace $\mathcal{W} \subset \mathcal{S}$ for the robot.*

At a code level, this requires that the user identify $s^{tar}$ in the environment definition, provide a means for sampling goal states, and select and operating region for the robot (such as the area over a table top). These pieces of information place minimal burden on the user, yet nonetheless contain rich information about problem structure which `OmniReset` exploits to design easily solvable RL problems.

**Key Problem Parameters:** `OmniReset` automatically generates the initial state distribution $s_0 \sim \rho$ with the right diversity which enables PPO to solve each of the tasks we consider with a single task agnostic reward function and zero tuning of algorithm hyper parameters.

## 3.2 AUTOMATICALLY GENERATING RL PROBLEMS WITH DIVERSE RESETS

Reinforcement learning fails to solve long-horizon manipulation tasks when training only encounters a narrow slice of potential object configurations or robot-object interactions. To prevent this collapse, `OmniReset` systematically expands coverage along two axes. First, we approximately cover the space of pathways along which the target object can be transported to the goal by densely resetting $s^{tar}$ on the tabletop, at random point in the air, and at and near the goal $\mathcal{G}$. This exposes the RL algorithm to difficult-to-discover success signals, and allows these signals to smoothly propagate throughout the state-space as the value function is updated. Next, we cover the different ways the robot can interact with $s^{tar}$ by resetting the arm in configurations where it is reaching towards $s^{tar}$, making contact with $s^{tar}$ from a wide variety of points, and with stable grasps on $s^{tar}$. This exposes the RL algorithm to the different ways the robot can interact with $s^{tar}$ to move it towards the goal. Altogether, this broad coverage enables the RL algorithm to discover high-value regions of the state space and the behaviors required to reach those states.

Specifically, we generate these resets as follows. First, we use the grasp sampler from Mittal et al. (2023) to calculate a set of 1000 feasible grasp points on the target object $s^{tar}$, as depicted in Figure 3a. Next, we generate a set of feasible offsets for $s^{tar}$ relative to the goal $\mathcal{G}$. This is accomplished by spawning $s^{tar}$ at $\mathcal{G}$, and then applying small random forces to dislodge the target from the goal, as in Tang et al. (2023). For example, for the insertion task in Figure 1, this process generates a continuum of relative configurations between the peg and the hole where the peg is only partially inserted. With these pre-computed quantities, we generate the following resets over the space of robot-object configurations:

**1) Reaching Resets:** $S^R \subset \mathcal{S}$ capture states where the robot is moving towards the target object. For our tabletop manipulation problems, this corresponds to resetting the target object at diverse poses on the table top, with the gripper spawned at random poses throughout the workspace $\mathcal{W}$.

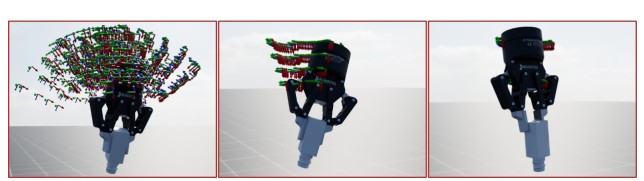

(a) **Grasp Sampling.** We display the grasp poses sampled for the table leg. The grasp sampling ranges are broad, moderate, and narrow from left-to-right. Our method uses the broad range.

**2) Near-Object Resets:** $S^{NO}$ also spawns $s^{tar}$ across the tabletop, but then resets the end effector to one of the pre-computed grasp points with a small random offset and randomly set the gripper to either be open or closed. This provides broad coverage over states where the robot is initiating contact with $s^{tar}$ from a wide distribution of directions, providing coverage over both non-prehensile interactions and the initiation of stable grasps.

**3) Stable Grasp Resets:** $S^G \subset S$ covers states where prehensile manipulation occurs and the robot has secured a stable grasp on $s^{tar}$. We spawn $s^{tar}$ randomly in the air throughout $\mathcal{W}$, then spawn the gripper at a feasible grasp point.

**4) Near-Goal Resets:** $S^{NG} \subset S$ provides dense coverage over near goal states where contact rich behavior such as insertion or twisting occur. We spawn $s^{tar}$ at one of the pre-computed offsets from the goal $G$, then spawn the gripper to be in contact with $s^{tar}$ as with the Near-Object resets.

Altogether, these different reset regions provide dense, approximate coverage over the space of pathways to the goal, without prescribing *a priori* which behaviors are needed to solve the task. Indeed, as we see from our wide range of examples (Fig.1) the RL algorithm is free to select completely different pathways through the state space when solving different tasks, utilizing the reset states which are useful while ignoring those which are not. For example, in the `Drawer Insertion` task, at convergence the RL algorithm does not obtain stable grasps on the drawer, but instead repeatedly flips the drawer and then pushes it into the cabinet. In contrast, for the `Leg Twisting` tasks, the

robot picks the table leg, pushes it against the table to obtain a more favorable grasp, and then twists the leg into the hole.

**Practical Implementation:** In practice it can be difficult to efficiently sample *feasible resets* which respect the contact constraints of the physics simulator. Sampling invalid initial conditions can lead to pathological and non-physical behavior which destabilize learning. Moreover, reset states must be sampled with minimal overhead to maximize GPU-parallelism. Thus, we first sample feasible resets during an offline phase, sampling proposed resets from the four regions defined above and rejecting invalid samples using a combination of collision checking and stepping the simulator for a few steps to allow object to settle. This yields four validated datasets corresponding to the four reset distributions described above: $D^R$, $D^{NO}$, $D^G$, and $D^{NG}$. During RL training, we sample from $\text{Uniform}(D)$ where $D = D^C \cup D^{NO} \cup D^G \cup D^{NG}$. We cache these resets on-GPU to ensure efficient sampling during training.

### 3.3 Algorithmic Decisions for RL Training

Next we discuss key algorithmic decisions which are required for PPO to leverage the diversity of resets states and scale to the complexity of tasks we consider.

**Task-Agnostic Reward Structure:** Leveraging the design choices described above, we use a simple, common reward function shared across all tasks:

$$r(s_t, a_t) = r_{\text{success}}(s_t) + r_{\text{dist}}(s_t) + r_{\text{reach}}(s_t) + r_{\text{smooth}}(s_t, a_t) + r_{\text{term}}(s_t). \tag{1}$$

Here, $r_{\text{success}}$ is a sparse binary reward indicating task completion, $r_{\text{dist}}$ encourages minimizing the distance of the target object $s^{tar}$ to the goal, $r_{\text{reach}}$ encourages the gripper to be near $s^{tar}$, $r_{\text{smooth}}$ penalizes large or rapidly changing actions, and $r_{\text{term}}$ penalizes unsafe or physically invalid states that trigger terminations. Importantly, this reward does not encode task-specific strategies as all components and weights are kept fixed across experiments. We find that this generic structure is sufficient for stable training across diverse manipulation tasks, and performance is largely insensitive to the precise weighting of individual terms. See the Appendix for additional details.

**Scaling Parallel Environments:** When combined with increasing reset diversity, we found that scaling the number of parallel environments used by PPO stabilized and accelerated learning. This enables `OmniReset` to obviate curricula and task dependent rewards, significantly reducing the amount of tuning which is required to solve a new task.

**Asymmetric Actor-Critic:** We asymmetric actor-critic approach (Pinto et al., 2017) for our learning architecture. The actor observations include a history of the five previous time-steps for the state of the robot, the poses of all objects in the scene, and the previous actions taken by the policy. The critic takes in these observations as well as additional privileged parameters of the environment. We found that conditioning the actor on larger observation spaces led to less stable training and led us to only provide this information to the critic.

**Generalized State-Dependent Exploration Noise:** We employ the policy noise parameterization from (Raffin et al., 2022). In short, gSDE has a separate prediction head which determines the gaussian exploration noise at each time-step and is conditioned on the features of the final layer of the policy network. This approach enables the actor to learn different temporally-correlated exploration strategies in different regions of the state-space, crucial for solving heterogeneous multi-stage tasks.

## 4 Simulation Experiments

We aim to address the following questions experimentally - (Q1) Does `OmniReset` outperform baselines and scale to tasks beyond current methods? (Q2) How do our key design decisions affect performance? (Q3) Can our experts be used to generate diverse data for sim-to-real transfer?

### 4.1 Task Descriptions

We use the following tasks to demonstrate the effectiveness of the `OmniReset` framework. The `Leg Twisting` tasks is a replication of the the `square_table` task from (Heo et al., 2023), and involves screwing in a single table leg. The `Drawer Insertion` task is based on the task by the same name from (Heo et al., 2023) and requires inserting a drawer into a dresser. The

`Peg Insertion` task requires inserting a peg into a hole. The `Cube Stacking` task requires stacking one cube on top of another with a desired orientation. The `Wall SLide` task requires non-prehensile motion to push a block up against a wall into a desired orientation. Finally, the `Cupcake Placement` task requires placing a cupcake on a plate at a desired orientation. For each task, we consider both `Hard` and `Easy` variants. For the `Hard` variants, the target object $s^{tar}$ is distributed on the table with $x - y$ coordinates in $(x, y) \in [-0.2, 0.2] \times [-0.15, 0, 15]$ and task-specific randomization over the goal position, whereas the `Easy` version of each task uses a highly restricted set of initial conditions with $(x, y) \in [0.1, 0.12] \times [0.1, 0.12]$ with a single fixed goal location. We refer the reader to our public code release for additional task specific parameters, but summarize each task below. Snapshots from the tasks are depicted in Figures 1 and 4

Finally, to fully demonstrate the utility of `OmniReset`, we solve the `Four Leg` task depicted in Figure 4 by $a$) training independent `OmniReset` policies to screw in each of the four legs then $b$) using a simple scripting policy to switch between the policies to complete the overall long-horizon task. The demonstrations highlights how `OmniReset` can be combined with high-level planning to push the horizon of tasks that can be solved to even greater extremes.

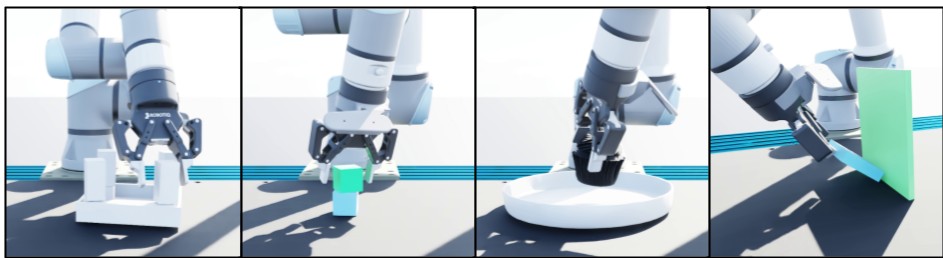

Figure 4: **Additional tasks.** Visualizations of the new manipulation tasks solved with OmniReset. From left to right: Four-Leg Table Assembly, Cube Stacking, Cupcake on Plate, Block Reorientation on Wall

## 4.2 SIMULATION EXPERT BASELINE COMPARISONS

We compare to the following baselines, which bootstrap learning with expert demonstrations, providing them more prior information about how to solve the task than `OmniReset`. For each of the methods, we supply 100 demonstration generated by the final `OmniReset` policy, effectively giving the baselines access to the optimal behavior. The initial conditions for these demonstrations are drawn from the reaching region $S^R$, which corresponds to the set of initial conditions the robot will encounter when solving the full task.

**1) BC-PPO:** We add a Behavior-Cloning (BC) loss to the PPO objective to construct a baseline emblematic of numerous works combining BC and RL objectives (Hester et al., 2018; Rajeswaran et al., 2017). When training this algorithm, the environment is always reset from the reaching region $S^R$, which reflects the 'standard' reset distribution typically used to solve such tasks.

**2) DeepMimic:** We use DeepMimic-style reward augmentation (Peng et al., 2018) on top of our generic rewards. During resets, a random demonstration is chosen, and the agent is reset from a random point along the demonstration and is given an auxiliary reward which provides bonuses for following the demonstration.

**3) Demo Curriculum:** This baseline is constructed in the spirit of method such as (Bauza et al., 2025) and uses a success-weighted autocurriculum to samlple reset states from the demonstrations. We use PPO as the base RL algorithm to ensure a fair comparison.

**Learning Curves and Success Rates:** We plot learning curves for the `Hard` variants of each task and the `Easy` variants of the `Peg Insertion`, `Leg Twisting`, and `Drawer Insertion` tasks in Figure 8. Success rates are reported for initial conditions sampled from the Reaching resets $S^R$, which correspond to states from which the robot must solve the full task. We see that `OmniReset` is able to consistently obtain high success rates on each of the tasks, substantially outperforming baselines. While the baselines are able to make some progress on the `Easy` variants of the tasks, they consistently struggle to scale to the wider distributions of initial conditions which defines the `Hard` variants. Figure 5 provides additional insight into the failure modes of baselines.

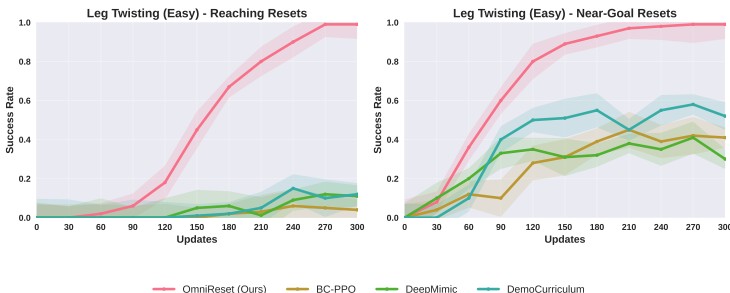

Figure 5: **Success Rates on Different Stages of Task.** We plot the success rates for the `Leg Twisting` task when starting from states that are in the Near-Goal region and also the Reaching Region of the state space. When evaluating these success rates, we sample resets from the demonstrations used by the baseline algorithms to ensure the resulting policies start from in-distribution states. We see that the baselines can achieve moderate success rates when starting close to the goal (Near Goal), but struggle to make meaningful progress on the full long-horizon task (captured by the reaching resets).

Here we plot success rates during training for initial conditions drawn from the demonstrations which fall into the Near-Goal and Reaching regions of the state-space. This plot demonstrates how the baselines are able to solve part of the task (i.e. when starting near the goal) but fail to scale to the full long-horizon task (i.e. starting from a reaching position).

**Robustness of Policies:** We conduct a robustness analysis on the learned policies (Fig 10). We sample an initial condition from one of the demonstrations, perturb the initial condition with forces of different magnitudes and report policy success from these perturbed initial conditions. We find that baseline performance quickly degrades under small perturbations, while the performance of `OmniReset` is barely affected under large perturbations.

We also analyze the ability of policies to solve the task from a wide range of initial conditions in the scatter plots in Figure (Fig 9). These plots show success rates over various initial conditions for both `OmniReset` and Demo Curriculum (the most successful baseline) on the `Leg Screw Easy` task. For each plot we show the success rate over 1000 sampled initial conditions from the full task distribution. This plot demonstrates how the baseline struggles to achieve achieve consistent success across the distribution of initial conditions it was trained on, while `OmniReset` achieves is able to succeed across the entire workspace.

### 4.3 ABLATING KEY DESIGN DECISIONS

We ablate 1) the number of parallel environments (and PPO batch size) and 2) the range of reset randomization used by `OmniReset` on the `Leg Twisting Hard` task. Figure 7 ablates the number of parallel environments and shows success rates during training from the four different reset distributions used by `OmniReset`. The reset distributions in Figure 7 are roughly ordered from left to right from the end of the task (Near-Goal) to the beginning of the task (Reaching). While runs with a smaller number of environments can make progress at solving the task from Near-Goal states, we observe that a large number of parallel environments are essential for scaling to the complexity of the full multi-stage task (i.e. from reaching states). Similarly, we see that increased diversity in the grasps used by `OmniReset` has a

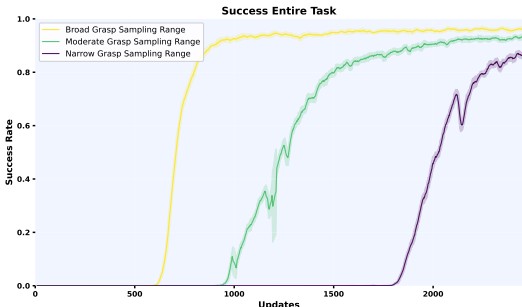

Figure 6: **Ablation on grasp sampling range.** For this ablation on the screwing task, we find that training RL on narrower grasp sampling ranges leads to worse sample efficiency and lower converged success rate.

substantial effect on sample efficiency, highlighting how densely covering the different modes of robot-object interactions is crucial for efficient and scalable RL training.

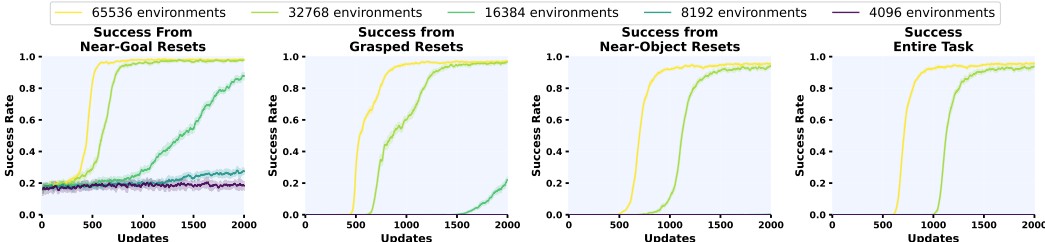

Figure 7: **Ablation on number of environments.** We plot the success rates over course of RL training using different number of environments. We find that the number of environments significantly impacts training performance.

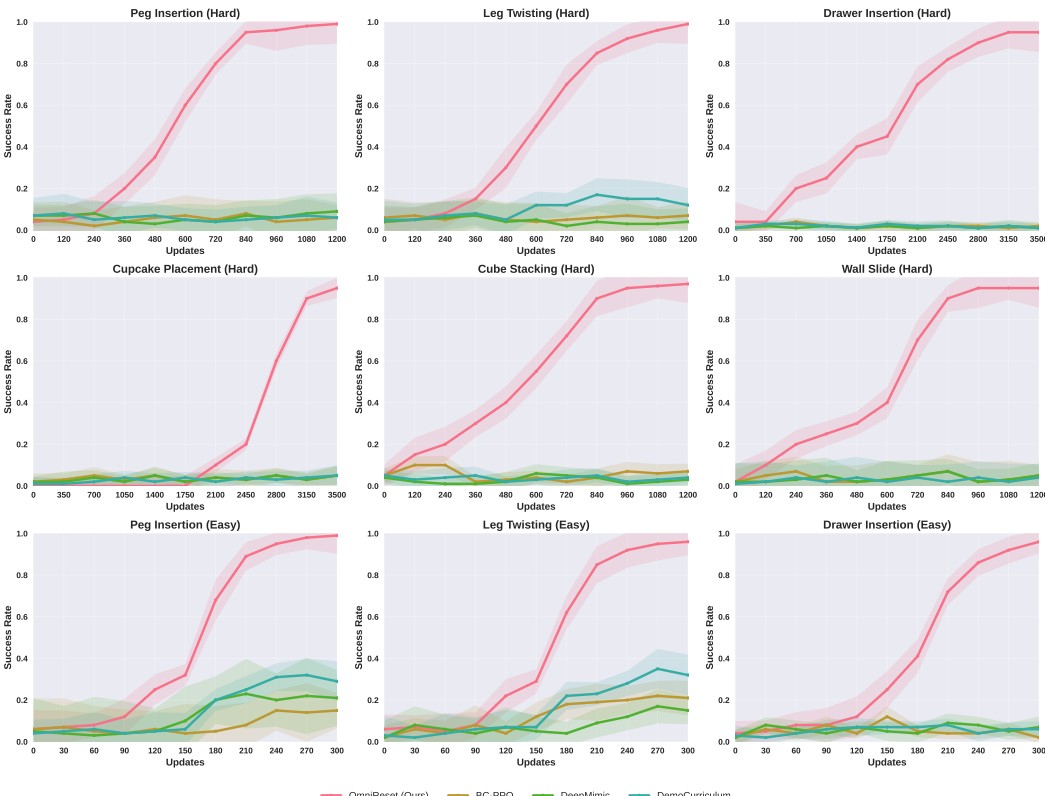

Figure 8: **Success rates during RL training.** We plot success rates over learning process for tasks described in Sec.4. We see that OmniReset scales to task where baselines struggle to make meaningful progress, especially with the wide range of initial conditions specified in the Hard variants of tasks.

## 5 DISTILLATION AND REAL-WORLD TRANSFER

We demonstrate the utility of our learned data-generation policies by distilling them into visuomotor policies deployable directly on hardware from RGB inputs. The robot observes $224 \times 224$ images from three cameras (front, side, wrist). Using the photorealistic rendering capabilities of IsaacLab (Mittal et al., 2023), we collect 10,000 expert rollouts with synchronized images and actions for standard student–teacher distillation Chen et al. (2021). The student policy uses an ImageNet-pretrained ResNet-18 encoder and a Gaussian MLP head conditioned on the five most recent observations.

**Visual randomization.** To mitigate the sim-to-real visual gap, we apply extensive domain randomization following DextrAH-G (Singh et al., 2025), varying lighting, backgrounds, object and robot appearance, and workspace textures. Camera extrinsics are calibrated to the real setup with additional pose and FOV jitter for robustness. We also apply standard image augmentations including color jitter, blur, grayscale, and noise.

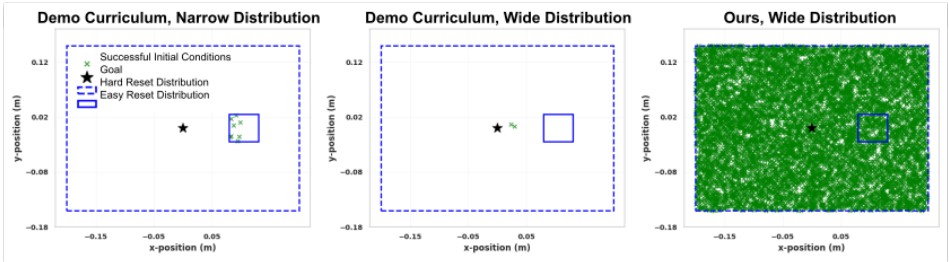

Figure 9: **Success states of RL policy.** For the screwing task, we plot the xy configurations from which RL polices trained with Demo Curriculum and `OmniReset` succeed when trained on the full reset distribution and on a narrow reset distribution. We find that `OmniReset` succeeds from a much broader range of initial conditions.

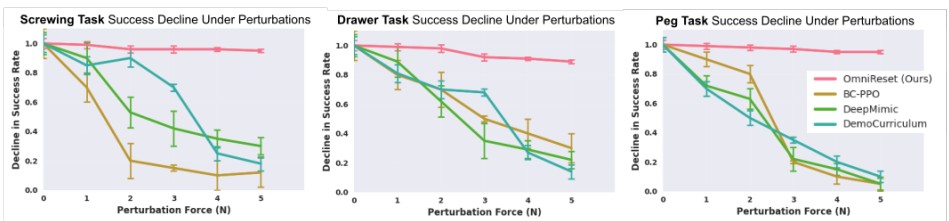

Figure 10: **Success rate over perturbations.** We plot the decline in success rate (measured by ratio of success rate between no perturbations and current level of perturbations). We find that `OmniReset` is robust to perturbations while performance of baselines drops significantly.

**Dynamics randomization.** To reduce the control gap, we calibrate kinematics to hardware and deploy the same task-space operational space controller in simulation and reality, with the policy predicting end-effector pose deltas that are converted to torques using identical Jacobian-based control. We perform system identification of key actuator parameters (e.g., friction and delay) to match real joint behavior, then randomize controller gains and physical parameters around identified values during RL training with a stability curriculum. We additionally apply an action-scaling curriculum to encourage smaller motions and randomize object mass and friction to promote contact robustness.

**Real-World Transfer.** We successfully deploy a distilled `OmniReset` policy for the `Peg Insertion Task`, as depicted in Figure 1. Evaluating our distilled RGB policy trained on 10,000 simulation trajectories zero-shot in real achieve a success rate of 25% from a wide range of initial conditions. We compare this to a Diffusion Policy Chi et al. (2025) trained on 100 demonstrations, which achieves only a 4% success rate. Qualitatively, the `OmniReset` visuomotor policy displays robust retrying behavior and is able to recover and successfully insert the peg even after initial failed attempts caused by the sim-to-real gap. Altogether, this demonstrates how `OmniReset` provides a foundation for scalable data generation for robust sim-to-real policies which are able to solve tasks from much wider ranges of initial conditions than existing methods.

## 6 CONCLUSION

In this work we presented `OmniReset`, a simple and scalable system for data generation in simulation for complex, dexterous tasks. The primary insight in `OmniReset` is showing that a diverse, minimally structured set of reset states paired with large-batch on-policy reinforcement learning in simulation can lead to the emergence of surprisingly complex dexterous behavior. We provide a general purpose recipe to instantiate data generators across a variety of manipulation tasks, and demonstrate both the efficacy of this paradigm in simulation and it's ability to bootstrap real world policy learning. The presented work is currently still single-task and object, making a multi-task extension in future work a promising direction of investigation. Furthermore, we are excited to extend `OmniReset` to higher-dimensional settings such as dexterous hand using existing techniques for generating grasps Xu et al. (2023). Furthermore, `OmniReset` opens up the possibility of more systematically studying scaling laws for policy learning.

# 7 REPRODUCIBILITY STATEMENT

We have made efforts to ensure reproducibility of our results by describing the steps of our data generation and training pipeline (Sec.3), our distillation and transfer pipeline (Sec.5), and our experimental results (Sec.4). Additional ablation studies are provided in the Appendix.

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
