# OpenReview forum: "Emergent Dexterity Via Diverse Resets and Large-Scale Reinforcement Learning"
_ICLR.cc/2026/Conference — ICLR 2026 Poster_

### Official Review · Reviewer_n5Rw · 2025-10-19

**Soundness:** 3
**Presentation:** 3
**Contribution:** 3
**Rating:** 4
**Confidence:** 3

**Summary:**

This paper focuses on general manipulation skill learning through reinforcement learning in simulation. The authors propose a method using large-scale generation of reset states based on user-specified points of interest, eliminating the need for complex reward or curriculum design while improving training efficiency and robustness. Experiments conducted in both simulation and the real world validate the effectiveness of the proposed approach.

**Strengths:**

1. This paper clearly identifies a critical problem in current reinforcement learning methods for learning general manipulation skills within simulation and proposes using large-scale, valuable reset states to constrain the exploration space.
2. The paper is clearly written, concise, and includes a comprehensive evaluation.

**Weaknesses:**

1. The current tasks are not long-horizon enough. For example, real-world assembly often involves multiple-part assembly. How should the points of interest be designed in such cases? For a chair with four legs, the set of points could be very large, making sampling potentially time-consuming.
2. The current experiments use only a robotic arm. How would the approach scale to a high-DoF dexterous hand? How can valid grasps be sampled?
3. For manipulation tasks that involve interaction with the environment, such as using a wall to adjust an object’s pose, current method seems can not effectively restrict the exploration space.
4. Since the points of interest may vary across objects, scaling to diverse objects might still require object-specific selection and significant user trial-and-error.

**Questions:**

1. How exactly are the goal configurations defined? What happens if defining them is difficult, for example, in the case of a dexterous hand?

---

> ### Author Response · Authors · 2025-11-25
>
> Dear Reviewer,
>
> Thank you very much for the detailed feedback on the paper! We have substantially revised our paper and updated our project website (https://sites.google.com/view/omnireset) to contain plots of the new results we provide, including four additional simulation tasks and new ablations.
>
> **Weaknesses:**
>
> 1.  **Task Horizon**: We have re-written the paper to carefully scope the class of tasks we consider (see `Task Scope` in Section 3). In short, we argue that OmniReset is ideally suited for learning single-step `macro-actions` such as assembling a single table leg, which require stitching together several lower-level behaviors. We argue that OmniReset does not need to directly scale to multi-step tasks, as it can be combined with high-level planners or VLMs which are better equipped to handle multi-step semantic reasoning. To illustrate this point, **we introduce a new task which requires twisting four legs into a table**. To accomplish this, we train an OmniReset policy capable of twisting a leg into a desired hole, and then implement a simple high-level planner which specifies the assembly order. To see the video of the trained RL policy on this task, please refer to the Additional Tasks section on our updated project website (https://sites.google.com/view/omnireset). Refer to Appendix A.2 in the paper for details on this additional task.
>
> 2. **Dexterous Hands**: We note that there are a number of mature techniques for automatically generating grasps for dexterous hands (https://arxiv.org/pdf/2503.08257v1, https://arxiv.org/pdf/2303.00938). We believe integrating these techniques into OmniReset is an exciting direction for future work, which we have indicated in Section 6 in the new draft. However, we view this as beyond the scope of the current paper, whose goal is to lay the conceptual foundations for generating valuable reset states for manipulation.
>
> 3. **Environment Interaction**: We added a new task which requires using **a wall to reorient a block to a desired orientation**. We did not modify OmniReset: the task uses the exact same goal-specification interface and the same reset-generation pipeline. A video of the learned behavior appears in the Additional Tasks section of our project website, and full task details are provided in Appendix A.2. Quantitatively, the success-rate curve for this task (teal curve in Figure 14 in Appendix A.1) rises to near-100% success by less than 1200 updates, matching the sample complexity of other non-assembly tasks.
>
> 4. **Scaling to Diverse Objects**: This is a fair critique and directly motivated us to eliminate the need for manually identifying “points of interest’’ on the objects. To do so, we introduced a simple modification to the procedure for generating `Near-Goal` resets that removes all task-specific geometric annotation (see `Near-Goal Resets` in section 3.1). Instead of marking specific task-relevant regions on the object, we place the full mesh of the target object at the goal pose and use its axis-aligned bounding box to define the Near-Goal region. Any state whose pre-perturbation bounding box overlaps with this goal-pose box is automatically classified as Near-Goal. We retrained all OmniReset policies using this heuristic and observed negligible change in sample complexity or final performance. **Importantly, this simplification means that the user only needs to provide (a) the goal configuration and (b) the target object mesh. This also makes the method immediately extensible to entirely new objects.** As long as the object has a mesh and a defined goal pose, OmniReset can automatically generate Near-Goal resets and train a policy with no additional annotation effort. We show this empirically by automatically generating Near-Goal states and training successful RL policies for diverse objects such as cubes, cupcakes, and the wall-reorientation block. As shown in the updated success-rate curves (Figure 14 and the per-task plots in Appendix A.2), all three tasks converge to near-100% success with 1250 updates. Videos of all trained policies are available in the Additional Tasks section of our updated project website.
>
> **Questions**:
> 1. **Goal Configurations**: Goal configurations are defined in terms of a set of goal poses for a target object specified by the user. However, we do not specify goal poses for the robot. Namely, OmniReset only requires relatively simple goal-specifications to solve complex tasks, and we do not believe it would be necessary to define goal poses for a hand (also see [1][2] for similar work on object-centric goal definitions). We have re-written Section 3 to make this point more clear.
>
> [1] Yunsheng Tian, et al. Assemble Them All: Physics-Based Planning for Generalizable Assembly by Disassembly, ACM TOG / SIGGRAPH Asia 2022.
>
> [2] Bingjie Tang, et al. AutoMate: Specialist and Generalist Assembly Policies over Diverse Geometries, RSS 2024.

---

### Official Review · Reviewer_anU6 · 2025-10-22

**Soundness:** 2
**Presentation:** 3
**Contribution:** 2
**Rating:** 4
**Confidence:** 3

**Summary:**

This paper proposes a method to generate diverse initial states to enable more efficient reinforcement learning of long-horizon manipulation tasks. This is demonstrated on three contact-rich manipulation tasks, compared with baseline methods that utilize demonstration data. Sim-to-real transfer is achieved via distillation and cotraining with real data.

**Strengths:**

The proposed idea is simple and well executed on the proposed tasks.

**Weaknesses:**

I think the paper lacks enough evaluation to turn this into a really strong paper, for example:

1. The number of tasks used is quite limited. Given the rich set of tasks introduced in furniture-bench, where two of the tasks are coming from, it would be nice to evaluate the method on more tasks and report how many tasks are successful. This can give an idea of the limitations of the proposed method and what follow-up work can do to improve the results.

2. Four different sources of reset distributions are proposed. However, this paper lacks ablations to verify how important those distributions are.

**Questions:**

1. The details of the baseline are missing, e.g., how many demonstrations are used?

2. Given the success of imitation learning using a diffusion policy with demonstration, it is not clear to me why a pure diffusion policy wouldn't work with demonstration. Can the authors provide some explanation?

---

> ### Author Response · Authors · 2025-11-25
>
> Dear Reviewer,
>
> Thank you very much for your informative questions and comments! We have substantially revised our paper and updated our project website (https://sites.google.com/view/omnireset) to contain plots of the new results we provide, including four additional simulation tasks and new ablations that remove individual reset distributions to quantify the contribution of each component.
>
> **Weaknesses:**
>
> 1.  **Range of Tasks**: We added 4 new manipulation tasks spanning diverse geometries, contact patterns, and horizons to better illustrate the breadth of behaviors OmniReset supports. 3 of the 4 are non-assembly tasks chosen to show that OmniReset generalizes to reorientation, stacking, and precise placement rather than being limited to assembly. Videos of the trained policies are available in the Additional Tasks section of our website, and Appendix A.2 provides task visuals and training curves. Across all 4 tasks, the RL policies achieve near-100% success within 1,250 PPO updates (less than 12 hours of training on 4xH200 GPUs), converging as quickly as our assembly tasks. Notably, Cube Stacking reaches near-perfect performance in 500 updates, illustrating that these tasks are straightforward to specify and solvable under OmniReset:
>
>     1. **Four-Leg Table Assembly**: assembling four legs into a table top, demonstrating reuse of a primitive OmniReset policy within a high-level sequence.
>     2. **Block Reorientation on Wall**: requires using a wall for non-prehensile reorientation (new category of interaction absent from the original submission).
>     3. **Cube Stacking**: multi-object spatial reasoning with orientation constraints.
>     4. **Cupcake on Plate**: placing a delicate object onto a small target area with precise, stable positioning.
>
> 2. **Reset distribution ablations**: We added ablations where each reset distribution is removed individually (see Figure 3 on the website or Figure 12 in Appendix A.1 in the paper). The results show clear degradation whenever a key reset source is missing. The Near-Goal and Grasped distributions are both critical for learning from the full initial-state distribution. Without Near-Goal states, the agent never reaches a success reward and remains at zero percent success. Without Grasped states, the policy can solve Near-Goal states but cannot solve earlier stages, including the initial distribution. The Near-Object distribution is not strictly required because the policy can eventually learn without it, but including it improves sample efficiency and reduces the amount of data needed to reach high performance.
>
> **Questions**:
> 1. **The details of the baseline are missing, e.g., how many demonstrations are used?**: We use 100 demonstrations which are generated by the optimal policy learned by OmniReset, which provides the baselines with high-quality trajectories for initialization. We have addressed this omission in the new draft (blue text in Section 5.2 in the paper).
> 2. **Given the success of imitation learning using a diffusion policy with demonstration, it is not clear to me why a pure diffusion policy wouldn't work with demonstration. Can the authors provide some explanation?** A diffusion policy trained on 100 real-world demonstrations sees only a narrow slice of the state space, so even small deviations lead to compounding error, quickly pushing the policy off the demonstrated manifold—consistent with the 0% success we observed under a broad initial-state distribution. OmniReset resolves this by automatically generating dense, diverse reset states in simulation, providing the broad coverage needed to prevent compounding error and produce robust policies that succeed from the full initial distribution.

---

> > ### Comment · Reviewer_anU6 · 2025-11-27
> > **update**
> >
> > thanks for the response. I have updated the score. It would still be nice to see scenarios where the proposed approach fails, or does the authors believe the propose framework can solve any reasonable task?

---

### Official Review · Reviewer_BWSv · 2025-10-29

**Soundness:** 3
**Presentation:** 3
**Contribution:** 3
**Rating:** 6
**Confidence:** 4

**Summary:**

This paper introduces OmniReset, a simple and scalable framework for training reinforcement learning policies for complex, long-horizon, and contact-rich manipulation tasks.

The framework's key insight is to cleverly bypass the difficult exploration problem in RL. It automatically generates a large-scale, diverse distribution of reset states that covers all "reasonable" points along the path to the goal. This allows complex, dexterous behaviors to
emerge naturally from standard large-scale RL optimization. OmniReset is designed to operate without human demonstrations, auto-curricula, or extensive task-specific reward/hyper-parameter engineering.

The method constructs its diverse initial state distribution(beta ρ) by composing four sub-datasets: Reaching Resets (D^R), Near-Object Resets (D^NO), StableGrasp Resets (D^G), and Near-Goal Resets (D^NG). This only requires minimal high-level user specifications: a set of goal configurations (G), a workspace bounding box (W), and a bounding box for near-goal/contact-rich states (NG).

The Experimental results show that OmniReset significantly outperforms demonstration-based baselines (BC-PPO, DeepMimic, Demo Curriculum) on Easy task variants and successfully scales to the Hard variants of Screw, Drawer, and Peg tasks, where baselines struggled to make meaningful progress. Furthermore, the resulting policies exhibit superior robustness to perturbations. Finally, the paper shows that policies trained in simulation can be successfully transferred to a physical robot through vision-based distillation and co-training with a small amount of real-world data.

**Strengths:**

1. Novelty in Solving Exploration: The main contribution is the proposal and validation of using a diverse, minimally structured set of resets as an alternative to complex auto-curricula or reliance on expert demonstrations to solve long-horizon RL exploration challenges.

2. Scalability and Performance: OmniReset successfully enables RL (PPO) to master complex, contact-rich tasks (e.g., Screw Hard) that were previously beyond the reach of existing techniques. Achieving success rates of over 97% on these benchmarks is a significant
performance breakthrough

3. Robustness: Policies trained with OmniReset show exceptional robustness to perturbations and succeed over a much broader range of initial conditions compared to baseline methods like Demo Curriculum.

4. Effective Sim-to-Real Pipeline: The work demonstrates a practical and high-value use case by distilling the learned expert policy into a visuomotor policy that achieves a 30% real-world success rate after co-training with only 100 real demonstrations, vastly outperforming a policy trained only on the real demonstrations (0% success).

**Weaknesses:**

1. Shifts Complexity from Algorithm to Task Configuration: While the paper minimizes algorithmic complexity, it shifts the burden to non-trivial task-specific configuration requirements, which are the basis of the diverse resets. This challenges the central claim of "minimal human input”

Near-Goal State Bounding Box (NG): Defining the bounding box for contact-rich states requires expert knowledge about the precise geometry and interaction points necessary to solve the task (e.g., the threads and corresponding hole for the Screw task). This step essentially encodes a critical part of the task solution as a manual configuration, introducing a significant, task-specific engineering step.

Pre-computed Grasps: The method relies on a pre-computed dataset of feasible grasps from a sophisticated grasp sampler. This is a powerful but unstated dependency. For novel objects or robot hands where such a tool is unavailable, generating this grasp dataset would be a substantial engineering effort in itself, acting as a form of implicit prior knowledge that is not universally available.

2. Uncertain Generalizability Beyond Assembly Tasks: The effectiveness of the proposed four-part reset strategy (R,NO,G,NG) is demonstrated on three tasks that, while complex, all fall within the category of rigid object assembly and insertion. The framework's generalizability to manipulation paradigms with different structures is unproven

**Questions:**

While the authors achieve strong results on grasping and insertion, can this design generalize to other tasks, and can you quantify the labeling effort (annotation cost) required to design the task-specific annotations?

---

> ### Author Response · Authors · 2025-11-25
>
> Dear Reviewer,
>
> Thank you very much for your informative comments! We have substantially revised our paper and updated our project website (https://sites.google.com/view/omnireset) to contain plots of the new results we provide, including 4 new simulation tasks and additional ablations.
>
> **Weaknesses:**
>
> 1. **Specification Complexity**:
>
>     1. **Near-Goal States**: We agree that defining bounding boxes requires expert knowledge, so we **removed this requirement entirely**. Rather than manually identifying task-relevant object regions, we derive the Near-Goal set directly from object geometry. We place the full mesh of the target object at the goal pose and compute its axis-aligned bounding box. A candidate reset state is labeled as near-goal if, before any perturbations are applied, the object’s current bounding box overlaps with this goal-pose bounding box. This yields a simple, automatic, and geometry-based rule for identifying near-goal states without requiring any task-specific heuristics. We re-ran all original and new tasks with this simplified definition and observed no meaningful change in sample complexity or performance. The updated learning curves in Appendix A.1 (Figures 9-14) show virtually identical convergence rates and final success rates under the new heuristic. Section 3 has been revised to highlight this simplification, which reduces user input to specifying only (a) the set of goal object poses and (b) the target object (see blue text for specific changes).
>     2. **Grasp Sampler**: We agree it is important to clarify that the grasp sampler is an important dependency for OmniReset. Fortunately, modern grasp samplers are mature and operate fully **off-the-shelf** for new objects, requiring no task-specific tuning. In our experiments, we **did not modify the grasp-sampling settings at all** across any of the tasks or objects. This demonstrates that the grasping component is general infrastructure rather than a source of per-task engineering burden. Furthermore, we note that reliable grasp samplers also exist for robot hands: https://arxiv.org/pdf/2303.00938. We have added clarifications in blue text in Section 3.1 to the paper to prevent the impression that grasp sampling requires per-task annotation.
>
> 2. **Range of Tasks**: We added 4 new manipulation tasks spanning diverse geometries, contact patterns, and horizons to better illustrate the breadth of behaviors OmniReset supports. 3 of the 4 are non-assembly tasks chosen to show that OmniReset generalizes to reorientation, stacking, and precise placement rather than being limited to assembly. Videos of the trained policies are available in the Additional Tasks section of our website, and Appendix A.2 provides task visuals and training curves. Across all 4 tasks, the RL policies achieve near-100% success within 1,250 PPO updates (less than 12 hours of training on 4xH200 GPUs), converging as quickly as our assembly tasks. Notably, Cube Stacking reaches near-perfect performance in 500 updates, illustrating that these tasks are straightforward to specify and solvable under OmniReset:
>
>     1. **Four-Leg Table Assembly**: assembling four legs into a table top, demonstrating reuse of a primitive OmniReset policy within a high-level sequence.
>     2. **Block Reorientation on Wall**: requires using a wall for non-prehensile reorientation (new category of interaction absent from the original submission).
>     3. **Cube Stacking**: multi-object spatial reasoning with orientation constraints.
>     4. **Cupcake on Plate**: placing a delicate object onto a small target area with precise, stable positioning.
>
> **Questions**:
> 1. **More Tasks and Labeling Effort**: We agree that the original task set was not broad enough to fully characterize the strengths of OmniReset. Based on this suggestion, we expanded the evaluation from 3 to 7 tasks and deliberately included non-assembly tasks to demonstrate that OmniReset generalizes beyond grasping and insertion. The only task-specific inputs a user must provide are (i) the success offset—the relative pose between the two objects at completion—and (ii) the bottom offset, which specifies the transform from the object’s origin to its physical bottom. Both values can be obtained in a few minutes using standard 3D modeling software such as Blender, keeping annotation cost minimal. Preprocessing the USD assets—verifying origins, correcting scales, placing objects into the success pose, and extracting both the success and bottom offsets—takes roughly 30 minutes of human effort. Once these inputs are set, generating the reset states is fully automated and requires approximately 2 hours of compute time.

---

> > ### Comment · Reviewer_BWSv · 2025-11-27
> >
> > The authors have largely addressed my concerns, and I will raise my score accordingly.

---

### Official Review · Reviewer_DGvL · 2025-11-01

**Soundness:** 3
**Presentation:** 2
**Contribution:** 3
**Rating:** 6
**Confidence:** 4

**Summary:**

This paper presents a simple method to enable RL to learn long-horizon manipulation tasks. The key idea is to initialize the scene in a diverse set of initial states relevant to the task. The authors argue that "attempting to cover this space of behaviors may initially
appear intractable," but "the space of ‘reasonable’ manipulation behaviors is actually surprisingly small". To this end, the authors propose resetting to reaching, near object, stable grasp, and near goal. A user aids the process by providing near goal states. The user needs to provide a set of goal configurations, a workspace for the robot, and a set of near goal states, including contact-rich and goal states.

The authors show results in 3 challenging versions of manipulation tasks - drawer, screw, and peg in simulation. Additionally, the authors show the transfer of the screw task to the real world. To achieve real robot results, a state-based policy is distilled into a vision policy. Additionally, the authors found that it's necessary to co-train this distilled policy with a small amount of real robot data.

**Strengths:**

1. Impressive long-horizon manipulation task
2. Empirically, significant improvement over baseline

**Weaknesses:**

1. Only one task is shown on the real robot. What was the reason for not showing real-world results on all 3 tasks?
2. The authors claim "OmniReset automatically generates these resets with minimal human input". In that regard, I would expect to see results on more tasks.

**Minor**
1. Line 186: "We will let $s ∈ S$ denote the state space" -> "We will let $s ∈ S$ denote the state".
2. "transition dynamics of the simulator by $s′ ∼ P(·|s)$" -> "next state sampled from the transition dynamics of the simulator by $s′ ∼ c. P(·|s)$"
3. "$a ∼ π(·|s)$ to denote control policies" -> "$a ∼ π(·|s)$ to denote action sampled fromm control policy"
4. Line 211: Suggestion $J(π) = E_{s_0∼ρ,a∼\pi}[Σ_{t=0}^∞ γ^t r(s_t, a_t)]$ and remove "expection is alos taken w.r.t. the actions..."
5. In Figure 2 caption & on line 342: space after OmniReset
6. Line 359: Grammar "Both setting the orientation"

**Questions:**

1. Line 372: "standard reset distributions $ρ^S$ described above" - what exactly is it, and which line is it defined in?
2. What happens if we initialize around demo states with added perturbations? With enough perturbations, do you expect the state distribution to be sufficient and work similarly to the proposed method?
3. Since the core idea is having sufficient reset coverage in relevant states, is it possible to gather them without human assistance? E.g., via disassembly and perturbations around it?
4. How were the 100 real-world demos collected?
5. Several related complementary works are cited in the Related Work section. What happens when Omni Reset is combined with some of them?

---

> ### Author Response · Authors · 2025-11-25
>
> Dear Reviewer,
>
> Thank you very much for your informative comments and helpful clarifying questions! We have substantially revised our paper and updated our project website (https://sites.google.com/view/omnireset) to contain plots of the new results we provide, including 4 new simulation tasks and additional ablations on using demonstrations for resets, rewards, or behavior cloning.
>
> **Weaknesses:**
>
> 1. **Range of Tasks**: We added 4 new manipulation tasks spanning diverse geometries, contact patterns, and horizons to better illustrate the breadth of behaviors OmniReset supports. 3 of the 4 are non-assembly tasks chosen to show that OmniReset generalizes to reorientation, stacking, and precise placement rather than being limited to assembly. Videos of the trained policies are available in the Additional Tasks section of our website, and Appendix A.2 provides task visuals and training curves. Across all 4 tasks, the RL policies achieve near-100% success within 1,250 PPO updates (less than 12 hours of training on 4xH200 GPUs), converging as quickly as our assembly tasks. Notably, Cube Stacking reaches near-perfect performance in 500 updates, illustrating that these tasks are straightforward to specify and solvable under OmniReset:
>
>     1. **Four-Leg Table Assembly**: assembling four legs into a table top, demonstrating reuse of a primitive OmniReset policy within a high-level sequence.
>     2. **Block Reorientation on Wall**: requires using a wall for non-prehensile reorientation (new category of interaction absent from the original submission).
>     3. **Cube Stacking**: multi-object spatial reasoning with orientation constraints.
>     4. **Cupcake on Plate**: placing a delicate object onto a small target area with precise, stable positioning.
>
> 2. **Real-World Results**: The focus of this paper is on the data-generation method in simulation, so full real-world distillation across all tasks was outside the immediate scope. That said, we agree that demonstrating all 3 tasks on the real robot would further highlight the utility of the generated data. We are currently running real-world distillation experiments for the remaining tasks and will include those results in an updated version soon.
>
> **Minor**: Thank you for catching these issues! We have addressed all points in the new draft.
>
> **Questions**:
> 1. **What is "$\rho^S$"?** $\rho^S$ is a uniform distribution over the reaching dataset $D^R$. We refer to this as the `standard` reset distribution, since it corresponds to the state from which we ultimately want to solve each task (i.e. objects are randomly distributed on the table and the gripper is starting at a random point in the workspace). We have properly defined this notation in the new draft (see equation 2).
> 2. **What if we initialize around perturbed demo states?** We ran initial experiments exploring this idea (Fig 1 on the website; Fig 5 in the paper). We do this by a) initializing at demo states and b) applying small random actions to the robot and adding small force perturbations to the objects to mine new reset states. We are seeing only a small boost in performance to the baseline algorithms. We hypothesize that, while perturbations to demos can provide additional *local coverage* over the state-space, it does not effectively place the *global coverage* OmniReset naturally provides.
> 3. **Can resets be gathered with disassembly and perturbations?** This is what OmniReset already does. OmniReset requires only the user to input a success-relative pose between the two objects. Given this specification, the entire reset-state generation process is fully automated: the system performs disassembly with force perturbations and creates a set of near-goal, grasped, and other reset states. This shows that broad reset coverage can be achieved automatically once the user provides this goal specification. We have updated Sec 3.1 in blue to make it clear how reset states are automatically generated.
> 4. **How were the 100 real-world demos collected?** They were collected using a GELLO teleoperation setup [1], which is a leader-follower puppeting system. For each demo, we randomized the object’s initial pose broadly across the table surface—sometimes starting already in the gripper, other times placed loosely on the table—to ensure diverse starting conditions.
> 5. **What happens when OmniReset is combined with related approaches?** We investigate whether adding resets from demos (like in several baselines) can provide an additional boost to OmniReset (see Fig 2 on the website or Fig 11 in Appendix A.1 in the paper). For this particular approach, we see only a minimal change in performance compared to OmniReset alone. We believe further investigation of how different exploration techniques can be combined with OmniReset to be an extremely interesting direction for future work.
>
> [1] Philipp Wu, et al. GELLO: A General, Low-Cost, and Intuitive Teleoperation Framework for Robot Manipulators, IROS 2024.

---

### Author Response · Authors · 2025-11-25

We thank all reviewers for their thoughtful feedback. We have substantially revised the paper and have also updated our project website (https://sites.google.com/view/omnireset) with videos of the new tasks and ablations. We highlight changes we made in the paper in blue. Below is a summary of the major updates we have made in response to the reviewers’ comments:

**Additional Tasks**: Each of the reviewers asked for additional tasks to verify the scalability of our approach. Specific requests include more task diversity, longer-horizon tasks, and tasks where interacting with the environment is necessary. To address these concerns, we provide the following additional tasks:

- **Four Leg Table Assembly**: the robot picks up each of the four legs sequentially, then inserts and twists each one into its corresponding table socket.
- **Block on Wall**: the robot must use the wall to reorient a block up into a desired location.
- **Cube Stacking**: the robot must place one cube on top of another in a desired orientation.
- **Cupcake on Plate**: the robot must place a cupcake at a desired position on a plate.

Videos of the learned RL behavior for these tasks are posted on our project website. Refer to Appendix A.2 in the paper for additional images and training curves for these tasks.

**Additional Ablations and Baselines**: Reviewers DGvL01 and anU6 requested ablations assessing the importance of each reset distribution, as well as baselines that use demonstrations or perturbed demonstrations for resets, rewards, or behavior cloning. We have run all of these experiments and updated our project website and paper with the corresponding plots (See Figure 5 in Section 5 as well as Figures 11-12 in Appendix A.1 for details). These experiments show that adding expert-visited states or using alternative demonstration-heavy baselines such as BC-PPO or DeepMimic with demos or perturbed demos yields little improvement, indicating that OmniReset already provides the necessary learning signal. Removing individual reset sources, however, causes clear degradation, confirming that Near-Goal and Grasped resets are essential and that Near-Object resets improve sample efficiency.

**Burden on User**: Reviewers BWSv and n5Rw raised concerns about the amount of human effort needed to specify a task for OmniReset, specifically the Near-Goal specification, which requires identifying which parts of objects must come into contact to solve the task. To address this we:
- Remove this assumption entirely, by introducing a new, simple heuristic rather than relying on the previous assumption (see `Near-Goal Resets` in section 3.1). A reset state is classified as near-goal if, prior to applying any perturbations, the object’s bounding box still overlaps with its original bounding box. This provides an automatic and consistent way to determine near-goal states based solely on object geometry.
- Rewrote the methods section to make it clear that the user now only needs to specify a) the set of goal states and b) the target object to be manipulated (see blue text in section 3).
- Solved each of the tasks we considered with the simplified near-goal specification (see Figure 4b for updated near-goal specification).

**Scope of Method**: Reviewer n5Rw asks for clarification on the scope of OmniReset, and whether it can be applied, for example, to dexterous hands. We address this by:
- Highlighting the key conceptual ideas behind OmniReset which can be applied to other settings in future work (`Identifying Key Regions of the State-Space` in Section 3.1).
- Added discussion indicating how OmniReset can scale to dexterous manipulation in future work using existing dexterous grasp sampling techniques (see blue text in Section 6 for details).

We point reviewers to their review-specific responses for a more detailed discussion of their concerns.

---

### Author Response · Authors · 2025-12-03
**Response to New AC**

Here we outline how we have systematically addressed each of the main issues raised by the reviewers. We have substantially revised the paper to reflect this discussion and have also updated our project website (https://sites.google.com/view/omnireset) with videos of the new tasks and ablations. We highlight changes we made in the paper in blue. Please see our direct responses to reviewers for more detailed responses.

**Additional Tasks**: Each of the reviewers asked for additional tasks to verify the scalability of our approach. Specific requests include more task diversity, longer-horizon tasks, and tasks where interacting with the environment is necessary. To address these concerns, we provide the following additional tasks:
- **Four Leg Table Assembly**: the robot picks up each of the four legs sequentially, then inserts and twists each one into its corresponding table socket.
- **Block on Wall**: the robot must use the wall to reorient a block up into a desired location.
- **Cube Stacking**: the robot must place one cube on top of another in a desired orientation.
- **Cupcake on Plate**: the robot must place a cupcake at a desired position on a plate.

Videos of the learned RL behavior for these tasks are posted on our project website. Refer to Appendix A.2 in the paper for additional images and training curves for these tasks.

**Burden on User**: Reviewers BWSv and n5Rw raised concerns about the amount of human effort needed to specify a task for OmniReset, specifically the Near-Goal specification, which requires identifying which parts of objects must come into contact to solve the task. To address this we:
- **Remove this assumption entirely** by introducing a new, simple heuristic to automatically generate near-goal states which requires **no task-specific input from the user**. See `Near-Goal Resets` in Section 3.1 and responses to reviewers for more details.
- Rewrote the methods section to make it clear that the user now only needs to specify a) the set of goal states and b) the target object to be manipulated (see blue text in Section 3).
- Solved each of the tasks with the new heuristic.
In summary, after feedback from the reviewers, we simplified our method to require truly minimal input from the user.

**Scope of Method**: Reviewer n5Rw asks for clarification on the scope of OmniReset, and whether it can be applied, for example, to dexterous hands. We address this by:
* Highlighting the key conceptual ideas behind OmniReset which can be applied to other settings in future work (`Identifying Key Regions of the State-Space` in Section 3.1).
* Added discussion indicating how OmniReset can scale to dexterous manipulation in future work using existing dexterous grasp sampling techniques (see blue text in Section 6 for details).
With the seven tasks now included in the paper, we maintain that we have demonstrated OmniReset is widely applicable to many domains beyond just assembly.

**Additional Real World Experiments**: Reviewer DGvL01 asked why we demonstrated sim-to-real transfer for only a single task. We emphasize that our main contribution is a framework for behavior synthesis and large-scale data generation in simulation, not a novel contribution for sim-to-real transfer. In fact, our sim-to-real methodology is a simple co-training recipe taken directly from prior work. Our real-world experiments serve as a proof-of-concept demonstrating how our data generation method can be used in sim-to-real pipelines. But importantly, the focus is really on a new type of data generation method, not a new type of sim-to-real transfer method. We are in the process of performing additional transfer experiments, but we believe that this topic deserves its own in-depth study in a future paper.

**Additional Ablations and Baselines**: Reviewers DGvL01 and anU6 requested ablations assessing the importance of each reset distribution, as well as baselines that use demonstrations or perturbed demonstrations for resets, rewards, or behavior cloning. We have run all of these experiments and updated our project website and paper with the corresponding plots (See Figure 5 in Section 5 as well as Figures 11-12 in Appendix A.1 for details). These experiments show that adding expert-visited states or using alternative demonstration-heavy baselines such as BC-PPO or DeepMimic with demos or perturbed demos yields little improvement, indicating that OmniReset already provides the necessary learning signal. Removing individual reset sources, however, causes clear degradation, confirming that Near-Goal and Grasped resets are essential and that Near-Object resets improve sample efficiency.

Finally, we highlight that the two reviewers to respond (BWSv and anU6) both acknowledged that we addressed the main concerns that they raised and subsequently raised their scores.

---

### Meta-Review · Area_Chair_J89w · 2025-12-27

**Summary:**

The paper presents OmniReset which proposes a strategy for training RL methods to learn long-horizon manipulation tasks by generating a distribution of reset states that the agent can be reset to.  This provides the agent exposure to interesting states that may be challenging for exploration alone to reach.  The work focuses on manipulation tasks that can be broken down into sequence of steps of moving a single object to a goal location, and considers four types of reset states (reaching, near-object, grasped, near-goal).

Overall, reviewers indicated that the proposed method provided a novel way to solve exploration (BWSv,n5Rw), with good performance on long-horizon manipulation task (DGvL,BWSv) with robust policies (BWSv) and an effective sim-to-real pipeline (BWSv).  R-anU6 also noted that the "proposed idea is simple and well executed on the proposed tasks."

The main concerns from reviewers is the limited number of tasks addressed and how much human input is required.  There were also concerns regarding missing ablations, missing details and other writing issues (see "Reviewer Concerns" for detail).  The AC believes most of these were addressed (2 reviewers indicated willingness to increase their scores) with the introduction of additional tasks and experiments, a heuristic to generate near-goal states without input from the user, and other improvements to the paper.

The AC notes that there are some additional improvements to the presentation that the authors should include:
- Requirements.  A bit more information about what should be specified for the goal configuration and workspace (to allow for the reader to judge the complexity of specification).
- Figure 3: Please indicate which of the steps requires human specification
- Figure 5: the colors for "OmniReset" and "DemoCurriculum + Perturbations" is too similar.
- Figure 5,6,7: Figure resolution is very low.  Please include as PDF so text is sharp.
- Minor typos:
  - Please make sure \citep is used for proper parenthetical citations (e.g. L071, L072, L101, L228, L539, ...)
  - Use \citet for in text citations (e.g. L309, L398, L402, L428)
  - Space before citation (L122)
  - L365: "Visual Randomizations" => "Visual Randomizations:"
  - L390: "expirements" => "experiments"
  - L535: "it's" => "its"

The AC agrees with reviewer sentiments and find the work to propose a simple idea that could be useful to the community.  As reviewers did not express concerns about prior work already exploring this direction, the AC believe that this work can be of interest to the ICLR community and recommends acceptance.

**Reviewer Concerns:**

Concerns from reviewers include:
1. Limited number and complexity of tasks (DGvL, BWSv, anU6, n5Rw)
   - Evaluation on more tasks (DGvL, anU6)
   - More tasks should be shown on real robot (DGvL)
   - Generalizability to more complex tasks, with more complex interactions with environment, diverse objects (BWSv, n5Rw)
   - *Addressed* with the additional of four tasks (including multi-step tasks, and tasks that requires interacting with the environment) in the appendix, with robot videos on anonymous website.

2. Questionable claim that "OmniReset automatically generates these resets with minimal human input" as human assistance is required (DGvL, BWSv, n5Rw)
   - *Addressed.* In the initial submission, user input is required to specify the near-goal states.  In the revised draft, the authors introduced a heuristic to determine the near-goal states based on the bounding box of the object.

3. Missing ablations (DGvL, anU6)
- on perturbing demo states for reset distribution (DGvL)
- on different types of reset distribution (anU6)
- *Addressed* with addition of additional experiments

4. Writing issues (DGvL) and missing details (anU6)
- *Mostly addressed* by improved rewriting.

The authors provided an author response and revised manuscript that addressed most of the reviewer concerns.  For the last point, the AC note that there are still some parts that are unclear and should be improved.

**Reviewer Scores:**

Reviewers are mixed on this work with two marginal accepts (DGvL,BWSv) and two marginal rejects (anU6, n5Rw).  Authors provided a response and revisions, with two reviewers (BWSv, anU6) indicating they are willing to increase their score (late Nov 26, and Nov 27).

---

### Decision · Program_Chairs · 2026-01-26

Accept (Poster)